# Factors Predictive for Immunomodulatory Therapy Response and Survival in Patients with Hypersensitivity Pneumonitis—Retrospective Cohort Analysis

**DOI:** 10.3390/diagnostics12112767

**Published:** 2022-11-12

**Authors:** Katarzyna B. Lewandowska, Inga Barańska, Małgorzata Sobiecka, Piotr Radwan-Rohrenschef, Małgorzata Dybowska, Monika Franczuk, Adriana Roży, Agnieszka Skoczylas, Iwona Bestry, Jan Kuś, Witold Z. Tomkowski, Monika Szturmowicz

**Affiliations:** 11st Department of Lung Diseases, National Research Institute of Tuberculosis and Lung Diseases, 01-138 Warsaw, Poland; 2Department of Radiology, National Research Institute of Tuberculosis and Lung Diseases, 01-138 Warsaw, Poland; 3Department of Respiratory Physiopathology, National Research Institute of Tuberculosis and Lung Diseases, 01-138 Warsaw, Poland; 4Department of Genetics and Clinical Immunology, National Research Institute of Tuberculosis and Lung Diseases, 01-138 Warsaw, Poland; 5Department of Geriatrics, National Institute of Geriatrics, Rheumatology and Rehabilitation, 02-637 Warsaw, Poland

**Keywords:** hypersensitivity pneumonitis, lung fibrosis, pulmonary function tests, treatment

## Abstract

Hypersensitivity pneumonitis (HP) is one of the interstitial lung diseases with clearly established diagnostic criteria. Nevertheless, pharmacologic treatment recommendations are still lacking. Most specialists use steroids as first-line drugs, sometimes combined with an immunosuppressive agent. Aim: The aim of the present retrospective study was to establish predictive factors for treatment success and survival advantage in HP patients. Methods: We analyzed the short-term treatment outcome and overall survival in consecutive HP patients treated with prednisone alone or combined with azathioprine. Results: The study group consisted of 93 HP patients, 54 (58%) with fibrotic HP and 39 (42%) with non-fibrotic HP. Mean (± SD) VCmax % pred. and TL,co % pred. before treatment initiation were 81.5 (±20.8)% and 48.3 (±15.7)%, respectively. Mean relative VCmax and TL,co change after 3–6 months of therapy were 9.5 (±18.8)% and 21.4 (±35.2)%, respectively. The short-term treatment outcomes were improvement in 49 (53%) patients, stabilization in 16 (17%) patients, and progression in 28 (30%) patients. Among those with fibrotic HP, improvement was noted in 19 (35%) cases. Significant positive treatment outcome predictors were fever after antigen exposure, lymphocyte count in broncho-alveolar lavage fluid (BALF) exceeding 54%, RV/TLC > 120% pred., and ill-defined centrilobular nodules in high-resolution computed tomography (HRCT). An increased eosinophil count in BALF and fibrosis in HRCT were significant negative treatment outcome predictors. The presence of fibrosis in HRCT remained significant in a multivariate analysis. A positive response to treatment, as well as preserved baseline VCmax (% pred.) and TLC (% pred.), predicted longer survival, while fibrosis in HRCT was related to a worse prognosis. Conclusion: Immunomodulatory treatment may be effective in a significant proportion of patients with HP, including those with fibrotic changes in HRCT. Therefore, future trials are urgently needed to establish the role of immunosuppressive treatment in fibrotic HP.

## 1. Introduction

Hypersensitivity pneumonitis (HP) is one of the most prevalent interstitial lung diseases (ILDs), together with idiopathic pulmonary fibrosis (IPF), sarcoidosis, and connective tissue disease-related ILDs (CTD—ILDs) [1,2,3]. It results from exposure to inhaled organic dust, i.e., “inducers” or inorganic substances (haptens), small enough to enter the bronchiole and alveoli. In genetically predisposed patients, various types of immune response develop, resulting in the lungs’ granulomatous inflammation [3]. Both innate and adaptive immunity are involved [3]. T lymphocytes are the critical components of the immune reactions leading to the granulomas’ formation in HP patients. Still, immune complexes formed by immunoglobulins and antigens also play a role in HP pathogenesis [3]. Some patients with HP develop the progressive fibrosing phenotype, which worsens survival [3]. More than 200 inducing antigens were documented in the past, and many new possible “inducers” have been described recently [1,2,3].

Historically, three types of HP were distinguished: acute, subacute, and chronic, depending on the period of symptoms duration. In the newest guidelines for HP diagnosis, issued by the American Thoracic Society (ATS), the authors proposed recognizing the two types of HP based on the radiological features: non-fibrotic and fibrotic [1].

Diagnosis of HP is complex. Symptoms are unspecific (cough, fever, malaise, weight loss, progressive dyspnea, and exercise intolerance) and may mimic many other respiratory disorders. The most important is establishing the relationship between antigen exposure and symptoms. Nevertheless, the exposure cannot be identified in as much as half of patients with fibrotic HP [1,2,3].

Radiologic features of HP are well-established, although not pathognomonic. Chest X-ray reveals the presence of nodular or patchy infiltrates and fibrotic changes. In some patients, the chest X-ray may lack the signs of interstitial lung disease. The most important for HP diagnosis is high resolution computed tomography (HRCT) of the chest. In non-fibrotic HP, ground glass opacities, mosaic lung attenuation, and at least one sign of small airway disease, i.e., centrilobular nodules or air-trapping, are present [1]. In fibrotic HP reticulation, traction bronchiectasis and honeycombing are visible, in addition to the above-mentioned parenchymal and airway-related changes [1,4].

Broncho-alveolar lavage fluid (BALF) cellular analysis is helpful in the differential diagnosis of both fibrotic and non-fibrotic HP, as the increased percentage of lymphocytes may be present even in patients with advanced fibrotic disease [5].

The most effective intervention in HP patients remains antigen avoidance. The treatment regimens for HP are not clearly established.

The data in the literature concerning the influence of immunomodulatory treatment on survival in patients with fibrotic HP are scarce and conflicting [6,7,8,9,10].

## 2. Materials and Methods

### 2.1. The Aim of the Study

The aim of the present retrospective study was to establish predictive factors for treatment success and survival advantage in patients treated due to HP with steroids alone or in combination with azathioprine.

### 2.2. Regulatory Board Approval

The study was accepted by the Institutional Ethics Committee (KB-14/2019). Patients’ consent was waived by the Ethics Committee because of the study’s retrospective nature. All personal data were anonymized. The publication does not include any data or features enabling the identification of any individual patient in the analysis.

### 2.3. Study Group

The study group comprised consecutive patients diagnosed with HP in a single pulmonary department between 2005 and 2017 who received immunosuppressive therapy due to significant impairment of the pulmonary function test (PFT) parameters.

### 2.4. Principles of HP Diagnosis

The diagnosis of HP was established based on the combination of clinical, radiological, and/or histopathological features. Patients were diagnosed with HP if

Occupational or environmental exposure to organic antigens was established, and the symptoms of the disease were clearly related to this exposure, and/or positive precipitant immunoglobulins G (IgG) against avian or bacterial antigens were confirmed;

and 

2.A typical presentation of HP features in HRCT was described (i.e., mosaic lung attenuation, air-trapping, centrilobular nodules, ground glass opacities, upper and middle-lobe predominant fibrosis, peribronchovascular fibrosis, honeycombing) [3];

and

3.An increased percentage of lymphocytes exceeding 30% was present in BALF.

If the criteria 1 + 2 + 3 were not fulfilled, the HP diagnosis was confirmed by trans-bronchial, cryo-, or surgical lung biopsy.

The detailed description of diagnostic procedures used by our group has been published previously [11].

### 2.5. Treatment

The treatment regimens consisted of prednisone (average initial dose of 0.5 mg/kg/day gradually tapered) or prednisone (initial average dose of 0.5 mg/kg/day, gradually tapered) combined with azathioprine (100–150 mg/day). The choice between the two regimens depended on the doctor’s decision. Short-term treatment outcomes and toxicity were assessed after 3 to 6 months.

The arbitrary treatment response criteria based on PFT and chest X-ray results are presented in Table 1.

Baseline clinical symptoms (fever, weight loss, and time from symptoms onset to diagnosis), baseline and post-treatment pulmonary function tests (PFTs), radiological assays (plain chest X-ray—baseline and post-treatment, HRCT—baseline), BALF cellular count at diagnosis were analyzed. The clinical data were extracted from the hospital database and patients’ files.

Pulmonary function tests (spirometry and whole body plethysmography) were performed as routine measures in all patients with Master Screen Body/Diffusion (Jaeger, Germany, 2002) according to the European Respiratory Society (ERS)/ATS guidelines [12,13] and reported as absolute values and percentages of predictive values according to the ERS reference equations [14]. The transfer factor of the lungs for carbon monoxide (TL,co) was measured with a single breath method using helium gas as the marker. The results were presented as a percentage of predicted values with a correction to hemoglobin concentration [15].

A chest X-ray was performed routinely in all patients at baseline and after 3–6 months of treatment and were independently reviewed by two radiologists experienced in ILDs. Radiological response to the therapy in follow-up chest X-ray (improvement, stable, or worsening) was assessed based on lung density changes compared to the baseline examination. Worsening was defined as an increased lungs density in at least two lungs’ fields and improvement as a decreased lungs density in at least two lungs’ fields. Radiologically stable follow-up chest X-ray showed no significant differences in lungs’ density or density changes visible in only one lung field compared to the baseline examination.

Baseline chest CT scans were performed mostly using at least a 16-slice data acquisition scanner. In most patients, HRCT scans were performed; in some of them, additional expiratory scans were performed. Complete imaging of the lungs during inspiration and a slice thickness of 1.5 mm or less were required. The scans were independently reviewed in our institution by two radiologists experienced in ILDs, blinded to the clinical data of the patients. The presence of ground glass opacities, centrilobular nodules, mosaic lung attenuation, signs of lung fibrosis, and enlarged lymph nodes was assessed. Fibrotic lung changes were coded as score 0—no fibrosis, score 1—reticular abnormalities +/− traction bronchiectasis, and score 2—honeycombing. In patients with fibrosis scores of 1 or 2, the diagnosis of fibrotic HP was established; the patients with fibrosis scores of 0 comprised non-fibrotic HP.

BAL was performed according to the ATS guidelines, as described previously [5,16].

Survival time was calculated from the first day of treatment to death or the end of the observational period (i.e., 30 November 2019).

### 2.6. Statistical Analysis

Statistical analysis was performed within R software environment with the use of stat, plyr, PMCMRplus, PMCMR, ca, data.table, orsk, pROC, ggplot2, survival, and survminer packages. *p* values of <0.05 were considered statistically significant.

The values were presented as mean ± SD or median and CI. Between group comparison for continuous variables in two groups was assessed with Student’s *t*-test, the Cochrane and Cox test, or the Mann–Whitney U test, where appropriate. For three or more groups comparison, ANOVA (Friedman) or Kruskal–Wallis tests were used. For categorical variables comparison, Pearson’s χ^2^ test or the Fisher exact test were used depending on the groups’ sizes. Survival analysis was performed using Cox proportional hazards regression. Kaplan–Meier curves for the whole group and specified variables were drawn with median survival assessment, if appropriate. Cut-off values were established using receiver operating characteristics (ROC) curves.

## 3. Results

### 3.1. Baseline Characteristics of the Study Group

We analyzed the hospital database between 2005 and 2017 and found 171 patients with ICD-10 diagnosis codes: J67.1–J67.9. Forty-four patients were excluded from the study due to no baseline HRCT available for review. Out of the remaining 127 patients, 34 (31 with clinical data available) did not receive pharmacological treatment and 93 received immunotherapy.

Untreated patients differed significantly from the treated ones regarding baseline VC%pred (mean 100.5 ± 24.4% vs. 81.5 ± 20.8%, respectively, *p* < 0.0001) and TL,co%pred (71.9 ± 21.8% vs. 48.3 ± 15.7%, respectively, *p* < 0.0001).

The baseline characteristics of the study group are presented in Table 2. Fibrotic HP was recognized in 54 (58%) patients and non-fibrotic was recognized in 39 (42%). Patients with fibrotic disease differed significantly from non-fibrotic group regarding pulmonary function tests parameters and BALF cellular components. Seventy-six patients (82%) received corticosteroids (mean initial dose 37 mg/day, gradually tapered) and seventeen (18%) received corticosteroids and azathioprine (mean dose of AZA—139 mg/day).

The median time to assess the treatment response was 5.8 months (IQR: 3.1–9.1).

### 3.2. Treatment Outcomes

Most patients benefited from the treatment. Mean relative VC change in the whole group was 9.56 ± 18.8%, and mean relative TL,co change was 21.42 ± 35.1%.

Chest X-rays showed improvements in 37 patients, stable disease in 37, and progression in 19.

The treatment outcomes, assessed according to the algorithm based on a combination of VCmax, TL,co, and chest X-ray, were classified as improvement in 49 (53%) patients, stable disease in 16 (17%) patients, and progression in 28 (30%).

Progression despite treatment was noted in 24 (45%) patients with fibrotic HP compared to only 4 (10%) in the non-fibrotic group, *p* = 0.00069 (Pearson’s test) (Table 3).

### 3.3. Treatment Outcome Predictors

Positive treatment outcome predictors were fever occurring after the exposition to inducing antigen, weight loss, lymphocytes ratio in BALF higher than 54% (based on ROC analysis (AUC) 0.71, 95%CI: 0.598–0.71), increased RV/TLC (% pred.) > 120 (i.e., above the upper limit of normal value), and centrilobular nodules described in baseline HRCT (Table 4).

Factors predictive of negative response to therapy were increased percentage of eosinophils in BALF and lung fibrosis (score 1 or 2) in baseline HRCT (Table 4).

**Table 4 diagnostics-12-02767-t004:** Treatment outcome predictors (univariate logistic regression analysis).

Variable	OR	CI	*p*-Value
Sex	0.549	0.226–1.335	0.186
Age at dgn	0.981	0.943–1.020	0.338
Time from symptoms onset	0.997	0.989–1.005	0.442
Precipitins present	1.707	0.665–4.383	0.266
TLC %pred	1.017	0.996–1.038	0.113
VC % pred	1.003	0.982–1.025	0.787
TL,co % pred	0.974	0.945–1.003	0.080
**Fever**	**11.82**	**1.50–93.26**	**0.019**
**RV%TLC >120%pred**	**3.352**	**1.251–8.983**	**0.016**
**Lymphocytes in BALF > 53.55%**	**8.30**	**2.24–3.79**	**0.001**
**CT–centrilobular nodules**	**3.078**	**1.186–7.983**	**0.021**
**CT–fibrosis (any)**	**0.133**	**0.041–0.425**	**<0.001**
**Eosinophiles in BALF > 2.875%**	**0.23**	**0.08–0.67**	**0.0006**

In multivariate analysis, only the presence of fibrosis in HRCT remained a significant factor of negative treatment outcome, and eosinophils in BALF > 2.875% were a borderline significant factor (Table 5).

The therapy was generally well-tolerated. Treatment-related adverse events (TRAE) occurred in 59 patients (63%), but only one case (hepatotoxicity) led to the treatment discontinuation. The most frequent TRAE was the increase in body weight in 38 (40.8%) patients. Infection occurred in 14 (15%) patients, including 1 person with urosepsis; 1 with herpes-zoster; and the remaining 12 with mild respiratory tract infections, hyperglycemia/diabetes, and musculoskeletal disorders (i.e., myopathy/osteoporosis, with 6 (6.5%) each). Other TRAEs included acne (four patients), arterial hypertension (three patients), dyspepsia and arrhythmia (two patients each), deep venous thrombosis (one patient), and hepatotoxicity (one patient).

### 3.4. Survival

Twenty-eight patients died during the observation period. The median survival time for the whole group was 138.7 months (LCI—114 months, UCI—unavailable).

The survival prognoses according to fibrotic score and treatment results are illustrated in Figure 1 and Figure 2.

Fibrotic lung disease was combined with a significant worsening in survival (Figure 1). Median survival times in the patients with fibrosis scores of 0, 1, and 2 were over 170 months (median not reached), 104 months, and 88 months, respectively (*p* < 0.0001).

Response to treatment or stabilization during therapy was combined with a significant survival advantage (Figure 2). Median survival times in the patients who responded to treatment, who were stable, or who progressed were 178 months, 104 months, and 88 months, respectively (*p* = 0.00027).

Significant positive prognostic indicators were VCmax > 65% pred., TLC > 72.5% pred., and positive response to treatment. Significant negative prognostic indicators were fibrotic changes in HRCT (*p* < 0.001) (Table 6).

## 4. Discussion

Hypersensitivity pneumonitis is a rare interstitial lung disease, often diagnosed in the phase of lungs fibrosis. The diagnostic criteria of HP were lately well-established [1,3], but the treatment recommendations are still lacking. In our department, patients with long-lasting HP, apart from antigen avoidance policy, usually receive a trial treatment with steroids or steroids and azathioprine (as a steroid-sparing agent) according to the doctor’s discretion. We decided to analyze treatment outcomes and survival in the group of patients with HP diagnosed between 2005 and 2017.

HRCT revealed signs of lung fibrosis in 58% of patients at baseline. We found that lung fibrosis in HRCT was a predictor of unfavorable treatment outcome, regardless of fibrosis score (i.e., with or without honeycombing), both in univariate and multivariate analysis. A similar observation was made by De Sadeleer et al., who found that corticosteroid treatment showed some effect on FVC and TL,co in the group of non-fibrotic HP but no impact in fibrotic HP [6]. Of note, patients who received treatment for HP in the study mentioned above had more advanced disease than those not treated [6].

Although our patients with lung fibrosis were less likely to respond to therapy, we found that 19 of 54 (35%) patients with fibrotic HP experienced improvement, and a further 11 (20%) showed stabilization of the disease. Importantly, this was noted despite the fact that patients with fibrotic HP had worse baseline functional parameters than those without fibrosis. Similar results were shown in Ejima et al. regarding changes in FVC%pred. in the cohort of HP patients with mild fibrosis (i.e., without honeycombing) treated with corticosteroids for two years [9]. All subjects who received treatment experienced the improvement in FVC%pred., whereas untreated patients declined [9]. Another study by Tony et al. also showed post-treatment improvement in lung function and exercise performance in both non-fibrotic and fibrotic HP patients [10]. In fibrotic HP, the increase in FVC, FEV1, and oximetry before and after 6MWT and 6MWD were less expressed but also significant [10]. Morisset et al. retrospectively analyzed the effect of azathioprine or mycophenolate mofetil added to a small dose of prednisone in patients with HP and found significant improvements in TL, co, but not FVC after a year of treatment [17]. Similar results were observed by Fiddler et al.; in this retrospective group of 30 HP patients, slower decreases in both TL,co, and FVC were noted in treated patients. Still, only for TL,co, the change reached statistical significance [18]. We also found improvements in both VC and TL,co in our study group: TL,co increase was more pronounced compared to VC.

It is well known that even in fibrotic HP, some features of active inflammation (i.e., fever after exposure to antigen, elevated lymphocyte count in BALF, or nodules/ground glass opacities in HRCT scan) may persist. In our group, 21% of patients reported fever in anamnesis; in 82%, BALF lymphocytosis exceeded 30%; and in 45%, centrilobular nodules in HRCT were present. Those variables were positive predictors of the response to treatment, i.e., patients with fever had almost 12 times higher probability of improvement after treatment, those with BALF lymphocytosis > 54% had over eight times higher probability, and those with centrilobular nodules had 3 times higher probability. Lymphocytosis in BALF was predictive of a better response to therapy in the other study groups. Nevertheless, the optimal threshold was lower (20–28%) [7,19].

Additionally, we found that the increased eosinophil count in BALF was related to worse treatment response (a positive response was four times less probable if eosinophils comprised more than 2.9% of BALF cells). Additionally, an increased percentage of eosinophils showed borderline significance as a treatment failure predictor in the multivariate analysis. Data on the correlation between BALF eosinophil count and treatment response in patients with HP are scarce. A positive correlation between the BALF concentration of IL-4Rα, a cytokine that mediates eosinophil recruitment, and the eosinophil count in BALF in patients with progression of HP despite treatment was shown by Sterclova et al. [20]. Our data support these findings, suggesting that increased BALF eosinophil count may be an easy-to-detect and reliable indicator of treatment failure. Further prospective studies to support this hypothesis are needed.

Our data show the positive impact of increased baseline RV/TLC ratio (>120% pred.) on treatment outcome (more than three times higher probability of treatment response). An increased RV/TLC ratio indicates hyperinflation, which may be caused by inflammation of small airways. According to our knowledge, this is the first report in the literature concerning the positive predictive value of RV/TLC on treatment results in HP patients. This hypothesis requires further studies.

Our patients with HP had relatively good life expectancy; median survival in the whole group exceeded 12 years. This is comparable to other published cohorts: 9.2 years in the fibrotic subgroup of de Sadeleer’s cohort [6], 9.3 years in the patients with unidentified antigen exposure, and 18.2 years in those with identified antigen in Fernandez Perez’s cohort [21].

We found that lung fibrosis in HRCT was the most important predictor of death (HR 2.9). Patients with fibrosis score 2 (i.e., honeycombing) had the shortest median survival (88 months), whereas in the non-fibrotic patients, the median survival was not reached (over 170 months). This observation aligns with that of Salisbury et al., who found that patients with fibrotic HP had worse survival than those with the non-fibrotic disease. In those with honeycombing in HRCT, the survival was comparable to that of IPF patients [22]. The same negative influence of fibrosis on survival was observed in the study of Fernandez Perez et al., who found a 2.4 times increased likelihood of death in the group of patients with the signs of fibrosis in HRCT. In that group, nearly 90% were treated with steroids, and more than one-third were additionally treated with a steroid-sparing drug [21].

In our study group, improvement or stabilization during immunomodulatory treatment had a significant favorable influence on survival. According to our knowledge, this is the first study that analyzed the type of therapy response as a survival predictor in HP patients.

Contrary to our observation, De Sadeleer et al. presented the negative influence of therapy with steroids on survival [6]. The same result was observed by Ejima. Still, after matching the groups of treated and untreated patients according to the disease severity, the authors showed a positive effect of steroid treatment on survival [9].

The dose of steroids used in the treatment of HP may influence the treatment efficacy and morbidity related to side effects. Most authors used the initial prednisone dose of 0.5 mg/kg/day in monotherapy and slightly lower doses if combined with MMF or AZA [8,9,17,19]. Adegunsoye found higher toxicity of prednisone in the dose of 40 mg/day compared to the dose of 40–20 mg/day (tapered) in combination with immunosuppressive therapy, which resulted in worse survival of the monotherapy group, even after adjustment for age, sex, race, FVC%pred., TL,co% pred., and identified antigen [8]. TRAEs were infrequent and relatively mild in our group. We noticed only two serious TRAEs: urosepsis and herpes zoster.

### Study Limitations

Our study had some limitations.

First, it was a retrospective study; the treatment was determined at the physician’s discretion, not randomly.

Second, the time of treatment effect assessment was short, which could negatively influence the results, especially when diagnosing disease stabilization. On the other hand, such quick reassessment prevented patients with disease progression from continuing potentially toxic and ineffective drugs.

## 5. Conclusions

Our study on patients with HP treated with prednisone alone or in combination with AZA showed that both non-fibrotic and fibrotic HP patients might positively respond to the immunomodulatory treatment. Fever after antigen exposure, lymphocyte count in BALF exceeding 54%, RV/TLC > 120% pred. and presence of centrilobular nodules in HRCT were positive treatment outcome predictors. Increased eosinophil count in BALF and fibrosis in HRCT (regardless of fibrosis score) predicted negative treatment outcome, with the latter remaining significant in the multivariate analysis.

Positive response to treatment and preserved baseline VC% pred. and TLC% pred. predicted more prolonged survival in the whole group, whereas fibrosis in HRCT was related to a worse prognosis.

The presented data indicate that patients with HP may benefit from immunomodulatory treatment even if the fibrotic disease is diagnosed. Further studies are needed to establish the role of immunotherapy in the fibrotic HP treatment algorithm.

## Figures and Tables

**Figure 1 diagnostics-12-02767-f001:**
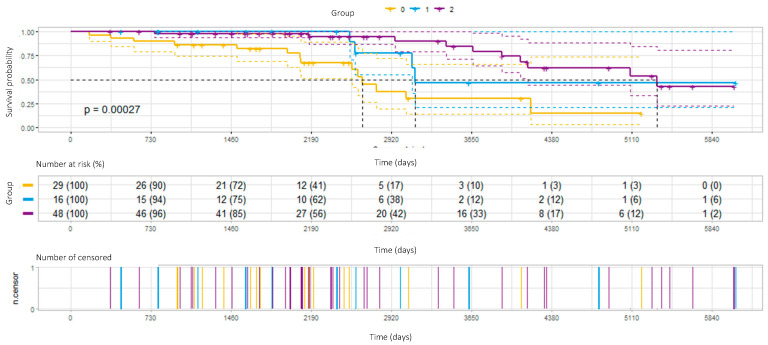
Kaplan–Meier survival curves in three different treatment outcome group: 0—progression, 1—stable disease, 2—improvement.

**Figure 2 diagnostics-12-02767-f002:**
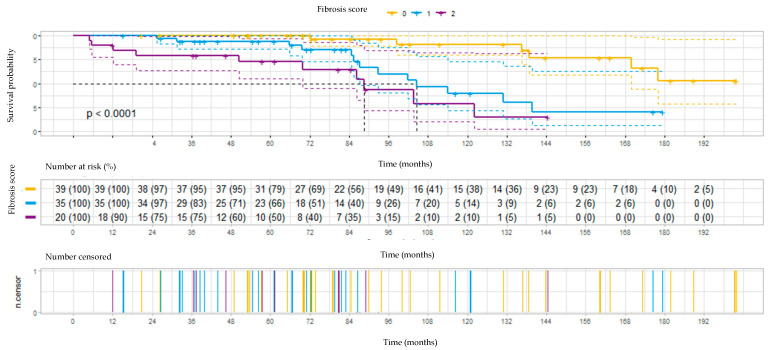
Kaplan–Meier survival curves according to the HRCT fibrosis score.

**Table 1 diagnostics-12-02767-t001:** Treatment outcome based on a combination of PFTs ad X-ray results.

	Treatment Outcome	Improvement	Stable Disease	Progression
Parameter				
**VC max ^1^**		Increase by 10% or more	Increase by < 10%	Any decrease
**OR**				
**TL,co ^2^**		Increase by 15% or more	Increase by < 15%	Any decrease
**AND**				
**Chest X-ray changes**		Improvement or stable	Improvement or stable	Worsening or stable

^1^—maximal vital capacity, ^2^—transfer factor of the lungs for carbon monoxide.

**Table 2 diagnostics-12-02767-t002:** Patients’ baseline characteristics.

Variable	Whole Group *N* = 93	Non-Fibrotic HP *N* = 39	Fibrotic HP *N* = 54	*p*-Value nf-HP vs. f-HP
Age at diagnosis (y), mean (± SD)	51.7 (±11.69)	49 (±12.81)	53.7 (±10.52)	0.0921
Male, N^o^ (%)	49 (53)	19 (48.7 *)	26 (48.15 ^#^)	0.9567
Ever smoker, N^o^ (%)	39 (41.5)	12 (30.77 *)	27 (50 ^#^)	0.0637
**Time from symptoms onset to diagnosis mean (± SD), mo**	**34 (± 40.5)**	**28.8 (± 62.92)**	**39.8 (±45.1)**	**0.0004**
Fever, N^o^ (%)	20 (21)	9 (23 *)	11 (20.3 ^#^)	0.7539
Antigen exposure, N^o^ (%)				
Poultry	32 (34)	17 (43.6 *)	15 (27.8 ^#^)	0.1271
Pigeons	19 (20)	8 (20.5 *)	11 (20.4 ^#^)	0.999
Parrots	4 (4)	1 (2.6 *)	3 (5.6 ^#^)	0.6368
**Hay/feed**	**43 (46)**	**26 (66.7 *)**	**17 (31.5 ^#^)**	**0.0015**
VC max (L), mean (± SD)	2.95 (±0.97)	3.14 (± 0.931)	2.80 (±0.989)	0.0729
VC max (% pred.), mean (± SD)	81.5 (±20.8)	86.5 (± 19.76)	78.3 (±20.34)	0.0622
**TLC (L), mean (± SD)**	**4.96 (±1.47)**	**5.58 (± 1.6)**	**4.48 (±1.16)**	**0.0006**
**TLC (% pred.), mean (± SD)**	**88.6 (±23.5)**	**99.7 (± 24.42)**	**78.5 (±16.84)**	**<0.0001**
**RV%TLC (% pred.), mean (± SD)**	**113.08 (±26.86)**	**122.4 (± 32.3)**	**106.1(±19.49)**	**0.0034**
**TLco (% pred.), mean (± SD)**	**48.3 (±15.7)**	**53.2 (± 15.35)**	**44.6 (±14.80)**	**0.0140**
Tiffenau index (%), mean (± SD)	80.2 (±8.23)	75.95 (± 13.97)	80.02 (±7.55)	0.2691
6MWD (m), mean (± SD)	485.7 (±106.3)	472.9 (± 116.4)	484.0 (±98.48)	0.7968
HRCT fibrosis any, N^o^ (%)	54 (58)	0	54 (100 ^#^)	
HRCT fibrosis—score 1, N^o^ (%)	35 (38)	0	35 (64.8 ^#^)	
HRCT fibrosis—score 2, N^o^ (%)	19 (20)	0	19 (35.2 ^#^)	
HRCT ground glass opacities, N^o^ (%)	79 (85)	35 (89.7 *)	45 (83.33 ^#^)	0.5466
HRCT centrilobular nodules, N^o^ (%)	42 (45)	21(53.85 *)	22(51.16 ^#^)	0.2920
BAL performed, N^o^ (%)	78 (83.9)	33 (84.6 *)	45 (85.18 ^#^)	0.999
BALF cells’ count (M), mean (± SD)	33.59 (±21.92)	41.47 (±28.1)	28.13 (±13.67)	0.0610
**BALF lymph (%), mean (± SD)**	**47.13 (±19.14)**	**59.98 (± 14.26)**	**38.48 (±17.10)**	**<0.0001**
BALF neut (%), mean (± SD)	6.1 (±6.03)	4.87 (±5.41)	6.97 (±6.34)	0.0924
**BALF eos (%), mean (± SD)**	**2.52 (±3.42)**	**1.68 (±2.34)**	**3.12 (±3.93)**	**0.0431**
TBLB performed, N^o^ (%)	29 (31)	14 (35.9 *)	15 (26.8 ^#^)	0.3722
SLB performed, N^o^ (%)	29 (31)	8 (20.5 *)	21 (38.9 ^#^)	0.0719
**CS monotherapy, N^o^ (%)**	**76 (82)**	**37 (94.9 *)**	**39 (72.2 ^#^)**	**0.006**
**CS + AZA treatment, N^o^ (%)**	**17 (18)**	**2 (5.1 *)**	**15 (27.8 ^#^)**	**0.006**

VC—vital capacity, TLC—total lung capacity, RV—residual volume, TL,co—transfer factor of the lungs for carbon monoxide, 6MWD—6 min walk distance, 6MWT—6 min walk test, 6MWT SpO_2_-1—6MWT oxygen saturation before the test, 6MWT SpO_2_-2—6MWT oxygen saturation after the test, HRCT—high resolution computed tomography, BAL—broncho—alveolar lavage, BALF—broncho-alveolar lavage fluid, TBLB—trans-bronchial lung biopsy, SLB—surgical lung biopsy, CS—corticosteroids, AZA—azathioprine, *—% of non-fibrotic group, ^#^—% of fibrotic group.

**Table 3 diagnostics-12-02767-t003:** Treatment response according to the presence of fibrosis in HRCT.

Characteristics	*N* (%)	Improvement *N* (%)	Stable Disease *N* (%)	Progression *N* (%)	*p*
Whole group	93 (100)	49 (53)	16 (17)	28 (30)	
Non-fibrotic HP	39 (100)	30 (77)	5 (13)	4 (10)	0.00069
Fibrotic HP	54 (100)	19 (35)	11 (20)	24 (45)	

**Table 5 diagnostics-12-02767-t005:** Treatment outcome predictors (multivariate logistic regression analysis).

Variable	OR	95%CI	*p*-Value
Fever	5.973	0.8116–125.0	0.1278
RV%TLC >120%pred	1.665	0.4462–6.404	0.4458
Lymphocytes in BALF >53.55%	2.048	0.3940–11.94	0.3958
CT–centrilobular nodules	1.737	0.5104–6.178	0.3794
**CT–fibrosis (any)**	**0.1489**	**0.0261–0.6552**	**0.0178**
Eosinophiles in BALF >2.875%	0.2669	0.0643–0.945	0.0527

**Table 6 diagnostics-12-02767-t006:** Predictors of survival (univariate Cox regression analysis).

Variable	OR	95%CI	*p*-Value
VC max > 65%	0.330	0.155–0.702	0.0025
TLC > 72.5%	0.298	0.140–0.633	0.002
Improvement after treatment	0.4511	0.293–0.693	<0.001
Fibrosis in HRCT	2.945	1.813–4.784	<0.001

## Data Availability

The data are available from the corresponding author upon request.

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
