# Peer review of "Factors Predictive for Immunomodulatory Therapy Response and Survival in Patients with Hypersensitivity Pneumonitis—Retrospective Cohort Analysis"

_diagnostics, 2022, doi:10.3390/diagnostics12112767_

Round 1

Reviewer 1 Report

This is a nicely presented retrospective study of patients with HP, which showed that fibrotic phenotype, BALF lymphocytosis and obstructive lung function are related with better prognosis.

I have however some questions to the authors:

1. The diagnostic criteria chosen by the authors are arbitrary. I wonder whether the use of the progressive pulmonary fibrosis criteria would be more appropriate, as suggested in the recently ATS/ERS/JRS/ALAT Guidelines for IPF and progressive pulmonary fibrosis (absolute VC decrease of 5%, DLCO decrease > 10% in a time period of two years). Is such an analysis possible? Otherwise, the authors should soundly explain why the chose the arbitrary criteria of table 1. 

2. Relevant to the previous comment, why did the authors decide to use chest x-rays as a progression criterion and not HRCT, since it is known that chest x-rays can underestimate the extent of disease in HP.

3. The associations reported between treatment response and baseline characteristics are univariate. A multivariate analysis would probably add power to the results.

4. A table comparing the baseline characteristics of fibrotic vs non-fibrotic HP would be useful, to reveal relevant differences.

5. The authors report the RV/TLC Ratio but not the Tiffeneau Index of their patients, both at baseline and after treatment. This is important, since a significant proportion of HP patients may present with an obstructive lung function pattern. In addition, the authors could add in the baseline characteristics how many patients had a concomitant diagnosis of COPD and how many patients were treated with inhalative therapy at baseline and in follow-up.

6.The authors report a mean time of 30 months from symptoms until diagnosis of HP: were the patients treated with oral corticosteroids or other medication for their symptoms in this time? 

Reviewer 2 Report

I appreciate the opportunity to review this work. i enjoy reading it and hope my comments can help authors to improve their work and publish it. i have 1 major and 1 minor comment.

1.- Major comment. the objective of this work is to retrospectively looked at factors that might predict who will response to therapy, as well as factors that might limit that response. patients were treated with two different strategies. so it is unclear to me if the positive predictive factors are driven by those receiving just steroids or those receiving double therapy. clarification on that point i feel it will be beneficial.

2.- minor comment: there is mention of table 5 but not table 5 on the manuscript.

Round 2

Reviewer 1 Report

I am satisfied with the revised version of the paper